# LINK PREDICTION ON TEXTUAL EDGE GRAPHS

## ABSTRACT

Textual-edge Graphs (TEGs), characterized by rich text annotations on edges, are increasingly significant in network science due to their ability to capture rich contextual information among entities. Existing works have proposed various edge-aware graph neural networks (GNNs), graph foundation models, or even let language models directly make predictions. However, they often fail to fully capture the contextualized semantics on edges and graph topology, respectively. This inadequacy is particularly evident in link prediction tasks that require a comprehensive understanding of graph topology and semantics between nodes. In this paper, we present a novel framework - LINK2DOC, designed especially for link prediction on TEGs. Specifically, we propose to summarize neighborhood information between node pairs as a human-written document to preserve both semantic and topology information. We also present a specialized GNN framework to process the multi-scaled interaction between target nodes in a stratified manner. Finally, a self-supervised learning model is utilized to enhance the GNN's text-understanding ability from language models. Empirical evaluations, including link prediction, edge classification, parameter analysis, runtime comparison, and ablation studies, on five real-world datasets demonstrate that LINK2DOC achieves generally better performance against existing edge-aware GNNs and language models in link predictions.

## 1 INTRODUCTION

In network science, the prevalence of networks with rich text on edges, also known as Textual-edge Graphs (TEGs) (Yang et al., 2015; Guo et al., 2019), increasingly become significant, as they encapsulate a wealth of relational and contextual information critical for diverse applications. The text associated with edges in networks can dramatically deepen our understanding of network dynamics and behavior. For instance, in a social media network, when a user responds to another's post, the reply not only creates a directed edge but also includes specific text that can reveal the sentiment, intent, or relationship between users. Similar cases can also be found in citation networks where text on edges is the exact reference quote. Both examples illustrate how text-rich edges are pivotal in accurately interpreting and leveraging networked data for advanced analytical purposes. In TEGs, link prediction is a unique yet open question due to the rich textual information embedded on edges.

Extensive works have been devoted to studying graphs with rich text, which can be classified into two categories, i.e., *Graph Neural Networks (GNN)*-based and *language model*-based. GNN-based works have extensively studied the topology connection between nodes and designed various variants. Specifically, works (Guo et al., 2019; Zhu et al., 2019; Gong & Cheng, 2019) typically compress the text embedded on edges to latent vectors by text encoders (e.g., Word2Vec (Mikolov et al., 2013) and BERT (Devlin et al., 2019)), and iteratively merge edge features with/without node features (Jiang et al., 2019; Yang & Li, 2020). The current most advanced edge-aware GNN (Jin et al., 2022) tries to refine the text encoder with the GNN training to obtain better representation. However, it still follows the neighbor aggregation way, which may not comprehensively consider overall semantics. On the other hand, owing to the strong text understanding ability of Large Language Models (LLMs) (Ling et al., 2023b), researchers (Chen et al., 2024; Fatemi et al., 2023; Ye et al., 2023; Yan et al., 2023) have directly used language models to solve graph mining tasks on textual graphs by designing various prompts to express or summarize the topology connection into natural language.

While these approaches have advanced the study of text-rich graphs, they tend to simplify the diverse text on edges, potentially losing crucial information necessary for tasks such as *link prediction*, where

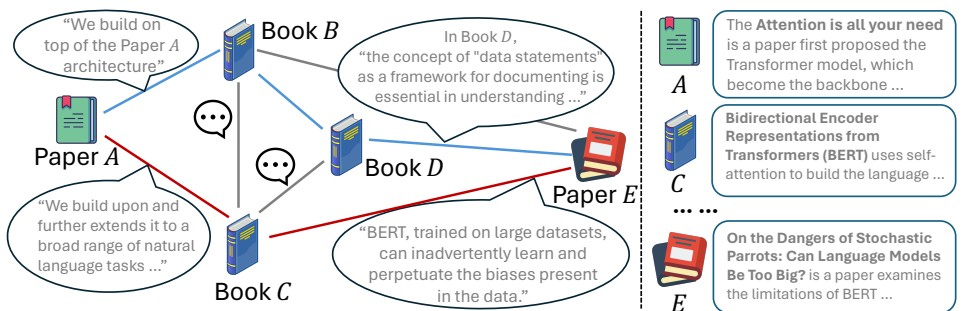

Figure 1: An example of textual-edge graphs: two books are connected by citation links. Predicting whether there'll be a citation between $A$ and $E$ needs to jointly consider both topology and semantic information embedded on nodes and their edges.

edge text is key to understanding the relationships within the graph. Both GNN-based and LLM-based methods may fall short of addressing the link prediction on TEGs due to two challenges, respectively.

**Challenge 1: Understand graph topology in language models.** For LLM-based approaches, existing works have worked on prompting LLMs by expressing or summarizing graph topology into text, but the graph topology is generally expressed in a linear shape, which may lead to a significant loss of graph-based structural information. For example, as shown in Figure 1, if we summarize the relation between Book $A$ and $E$ as Breadth-first Search, the dependencies along specific paths would be overlooked. This approach would not capture the multi-hop interactions and the rich contextual dependencies among nodes and edges. On the other hand, if we summarize all paths from $A$ to $E$, the same edge, such as the negative reference on edge $C$ to $E$, could appear in several paths. Representing this repeated edge in text multiple times leads to unnecessary duplication and potential noise, complicating the model's ability to discern the true nature of the relationships. If the graph is too large, potential overflow of the language model's context window limitation may also emerge.

**Challenge 2: Comprehensively consider context information on all connections.** For GNN-based methods, predicting links in TEGs requires a comprehensive examination of all potential paths connecting two nodes. Neighborhood aggregation approaches often focus on immediate neighbors and fail to account for the complex interactions that can occur in textual-edge graphs. As shown in Figure 1, there are many multi-hop paths with different intermediate nodes between $A$ and $B$, and each edge is annotated with different contexts, including user descriptions and user comments, describing their complex relations. It's already hard for existing GNN-based methods to use one latent vector to describe the complex context on edges. Moreover, user's preferences may also differ from one path to another (i.e., $A \rightarrow C \rightarrow E$ is negative while $A \rightarrow B \rightarrow D \rightarrow E$ is positive), creating a conflicting semantic landscape. However, GNN's neighbor aggregation would treat these paths uniformly, potentially diluting the sentiment difference in these paths.

**Present Work**. To effectively make link predictions on TEGs by jointly considering rich semantic information and graph topology, in this paper, we propose a novel representation learning framework, LINK2DOC, that transforms local connections between nodes into a coherent document for better-reflecting graph topology along with semantic information. To process rich textual information efficiently, we further propose a stratified representation learning framework that captures multi-scale interactions between target nodes. The crafted document further enhances GNNs to make link predictions in a contextualized way. The key contributions are summarized as follows:

- **Problem**. We formulate the problem of link prediction on textual-edge graphs and highlight the unique challenges of learning representations on textual-edge graphs for link predictions.

- **Method**. We propose an integrated framework to jointly consider topology and semantic information in textual-edge graphs, which consists of 1) coherent document composition to summarize semantic relations between node pairs in plain language; 2) a specialized Transition Graph Neural Network to process topology information between target nodes in a stratified manner; and 3) a self-supervised learning module to combine semantic understanding and topology processing ability for better link prediction on textual-edge graphs.

- **Experiment**. We empirically compare our method against existing state-of-the-art in four real-world datasets. Results have shown our proposed methods can elevate the performance of general GNNs and achieve competitive performance against edge-aware GNNs.

## 2 RELATED WORKS

**Edge-aware Graph Representation Learning.** Earlier research of Graph Neural Networks (GNNs) (Wu et al., 2020; Cui et al., 2023) tends to only focus on node features. Later on, research on heterogeneous graph representation learning (Yang et al., 2020) began to consider categorical information on edges. In text-attributed graphs, to more comprehensively utilize edge information during the network representation learning, some edge-aware GNNs (Zhu et al., 2019; Gong & Cheng, 2019; Jiang et al., 2019; Yang & Li, 2020) were proposed to consider edge text by designing various architectures (e.g., attention on edges, node and edge role switch, etc.). The recent state-of-the-art method EdgeFormer (Jin et al., 2022) involved pre-trained language models and proposed to better consider edge text by designing a cross-attention mechanism to merge node and edge representation in Transformer layers. However, these approaches still use the neighborhood aggregation way to obtain graph representation and cannot consider the local connections as a whole unit. Neighborhood aggregation may not always work especially when nodes are dissimilar (as shown in Figure 1) (Xie et al., 2020). Moreover, existing edge-aware GNNs tend to deliberately erase text on nodes and only explore the effect of edge information on various graph-related tasks (Jin et al., 2022), which lacks the flexibility to extend to other text-attributed scenarios.

**Language Modeling Augmented Graph Learning.** Large language models have been proven to have the ability to interpret graph-structured data (Jiang et al., 2023; Guo et al., 2023; Jin et al., 2023). In the past year, many works (Chen et al., 2023; 2024; Huang et al., 2023; Pan et al., 2024) have been proposed to prove LLMs have great potential (and even become state-of-the-art) to classify nodes in text-attributed graphs. However, how LLMs can better assist link prediction in text-attributed graphs is still an under-explored area, let alone the more complicated scenario of edge-attributed graphs that contain rich textual information on both edges and nodes. Existing works (Zou et al., 2023; Zhao et al., 2023; Li et al., 2023; Wen & Fang, 2023) have tried to distill implicit knowledge from LLMs to smaller GNN models for text-attributed graph tasks, but they still focus on learning good node embeddings. In edge-attributed graphs, text on edges cannot be uniformly processed in the same manner as node text, necessitating more specialized techniques that account for the unique semantics and structural roles of edge attributes in enhancing graph-based learning models.

## 3 LINK PREDICTION ON TEXTUAL-EDGE GRAPH

In this section, we begin by introducing key notations and formulating the problem of link prediction on Textual-edge Graphs. We then describe a novel way of constructing a transition document that summarizes the relationship between node pairs for link prediction. Finally, we provide an LLM-enhanced Graph Neural Network framework that learns the local topology and semantic information to retain both efficiency and efficacy.

### 3.1 PROBLEM FORMULATION

A Textual-edge graph (TEG) is a type of graph in which both nodes and edges contain free-form text descriptions. These descriptions provide detailed, contextual information about the relationships between nodes, enabling a richer representation of relational data than in traditional graphs.

**Definition 1** (Textual-edge Graphs). *A TEG $\mathcal{G} = (\mathcal{V}, \mathcal{E})$ is an undirected graph, which consists of a set of nodes $V$ and a set of edges $\mathcal{E} \subseteq \mathcal{V} \times \mathcal{V}$. Each node $v_i \in \mathcal{V}$ contains a textual description $d_i$, and each edge $e_{ij} \in \mathcal{E}$ also associates with free-form texts $d_{ij}$ describing the relation of $(v_i, v_j)$.*

In this work, we target at the *Link Prediction* task on TEGs, where we aim to predict the existence (or the label) of edges between pairs of nodes $(v_i, v_j) \notin \mathcal{E}$ based on the neighborhood information of $(v_i, v_j)$. Due to the rich edge text information, local edges in TEGs can inherently be represented by natural language sentences. For example, the connection $v_i \to e_{ij} \to v_i$ can be represented as "$d_i$ `is connected to` $d_j$ `via` $d_{ij}$", which directly describes the relation in plain text.

Compared to categorical-edge graphs that have edges labeled with simple, predefined categories, textual-edge graphs feature edges annotated with free-form text, which offer detailed and contextual relationship descriptions. Take Figure 1 again as an example, books are connected by textual edges, where text on edges consists of exact quotations that one book cites another. To predict whether $A$ and $E$ will have a citation link, we not only need to analyze semantics embedded within each edge's description, but different paths may also depict different semantic meanings due to the varying textual

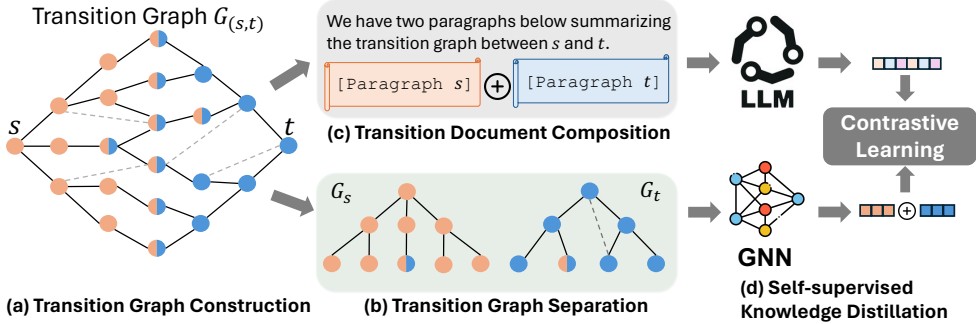

Figure 2: Overall framework of LLM-enhanced link prediction on Textual-edge graphs, where orange and blue nodes in $G_{(s,t)}$ belong to $s$'s and $t$'s local neighborhood (namely $G_s$ and $G_t$), respectively. Half blue and half orange nodes denote shared nodes between $G_s$ and $G_t$.

descriptions and types of relationships they represent. As noted in Figure 1, the Red Path contains negative reference from $A$ to $E$, while the Blue Path indicates another group of researchers endorse the research conducted by book $A$.

**Challenge.** The rich and complex text on edges makes the link prediction on TEGs not a trivial task, and there are two essential difficulties regulating existing *GNN-based* and *LLM-based* methods, respectively. For edge-aware GNN-based methods, directly combine and update each node $v_i$'s feature based on its neighbor's $\mathcal{N}_{v_i}(v_j)$ features as well as features of edges $e_{ij}$. However, neighborhood aggregation would fall short since semantics carried on each edge $d_{ij}$ needs to be viewed in the context of the whole connections from $s$ to $t$. For LLM-based methods, existing works tend to prompt language models by linearly summarizing graph topology, e.g., "$G_{(s,t)}$ contains $s, v_1, v_2, ..., v_n, t$ nodes, $v_1$ is connected to $v_2$ via $d_{12}$, $v_2$ is connected to $v_4$ via $d_{24}$, etc." This way may let LLMs fail to understand how information propagates from $s$ to $t$ and the contextual dependencies among nodes.

### 3.2 OVERALL ARCHITECTURE

In this work, we introduce LINK2DOC, a novel approach that leverages a self-supervised learning scheme to endow GNNs with text comprehension capabilities akin to those of LLMs. LINK2DOC is designed to preserve and synergize rich semantic information, topology information, and their interplay within TEGs for link prediction. We propose learning and aligning representations from two complementary perspectives: the *text view* and the *graph view*. The *text view*, termed *Text-of-Graph*, organizes the text associated with TEG's nodes in a way that reflects the graph's topology, forming a structured document that inherently captures logical and relational data. Conversely, the *graph view*, or *Graph-of-Text*, transforms the nodes and topology of TEGs into structured graph data. By employing pretrained language models (PLMs), the text view adeptly maintains textual integrity, while the graph view, processed through GNNs, ensures the retention of graph-specific characteristics. Aligning these views allows each representation to enrich the other, fostering a holistic understanding where textual nuances inform graph structures and vice versa.

Specifically, as noted in Figure 2 (a), to reduce the search space and to eliminate the noise from unrelated connections, we propose to formulate a $(s,t)$-transition graph containing all the possible routes through which $s$ could correlate to $t$ for link prediction.

**Definition 2** (Transition Graph). *For any pair of two entities $(s,t)$ in the Textual-edge Graph, all paths from $s$ to $t$ collectively form an $(s,t)$-transition graph, which is denoted by $G_{(s,t)}$. We use $n$ and $m$ to denote the number of nodes and edges in $G_{(s,t)}$, respectively. Figure 1 exemplifies an $(s,t)$-transition graph, where $s$ is Book A and $t$ is Book B. In practice, the length of paths can be upper-bounded by an integer $K$, which can usually be set as the diameter of the Textual-edge graph.*

The transition graph $G_{(s,t)}$, as well as all the text on edges, provide the necessary information needed to understand the relation between $s$ and $t$. Next, as shown in Figure 2(b), we separate two subgraphs: $G_s$ and $G_t$ from $G_{(s,t)}$ for preserving local neighborhood of $s$ and $t$, respectively. In Figure 2(c), to view the topology and semantic information of $G_{(s,t)}$ in a joint unit, we compose a manageable and coherent document that expresses $G_{(s,t)}$ as a human-written document. Finally, in Figure 2(d), we distill the text processing ability from large language models to graph neural networks for inference scalability while maintaining performance.

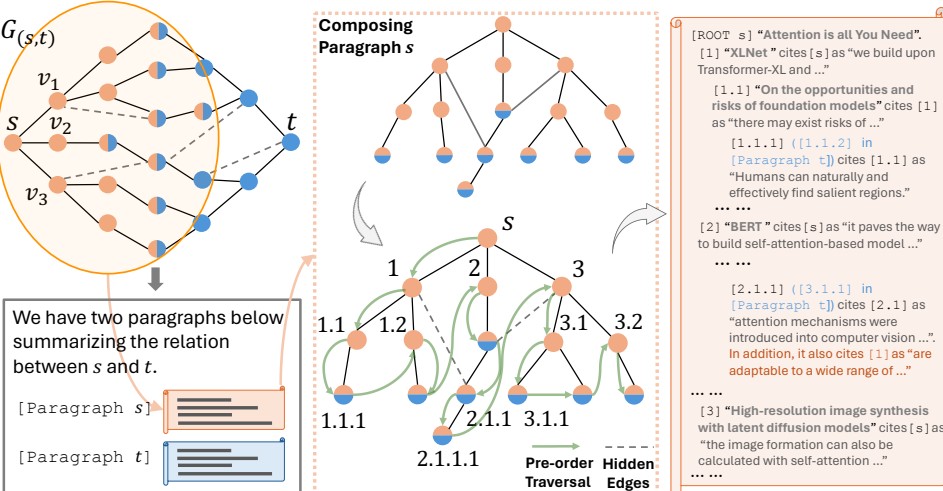

Figure 3: We first split $G_{(s,t)}$ into $G_s$ (nodes are marked with orange) corresponding to the local structure of $s$ ($G_t$ is omitted due to space limit). Commonly shared nodes are marked with half blue and half orange. We transform the local structure of $G_s$ into a paragraph that summarizes hierarchical relation with $s$ being the root. For better visibility, hidden edges are highlighted with orange, and commonly shared nodes are highlighted with blue.

## 3.3 TRANSITION DOCUMENT CONSTRUCTION

In this work, to address the first challenge, we need to summarize local relations in $G_{(s,t)}$ to comprehensively understand the relation between $(s, t)$. Since state-of-the-art LLMs are predominantly trained on human-written documents and books, in this work, we propose a novel way to express $G_{(s,t)}$ as a structured document $d_{(s,t)}$ (Lee et al., 1996), complete with an introduction, sections, subsections, and a conclusion, to let language models better understand the overall semantics between $s$ and $t$ with preserving topology. The overall process is illustrated in Figure 3, and the algorithm is summarized in Appendix 1 due to the space limit.

**Node-centric Paragraph Composition.** For source node $s$ and target node $t$, we first conduct Breadth-first Search (BFS) to extract their respective local structure with depth $L$, i.e., $G_s$ and $G_t$ from their transition graph $G_{(s,t)}$. As shown in Figure 3, nodes in $G_{(s,t)}$ that belong to $G_s$ are marked with orange, and nodes belonging to $G_t$ are marked with blue. The structural data (i.e., BFS tree) obtained is transformed into a textual paragraph aimed at both human readability and machine processability. Specifically, taking the BFS tree $G_s$ rooted at source node $s$ as an example, $s$ acts as the main subject of the paragraph's textual summary. We conduct pre-order traversal to go over all nodes in both trees. The first and subsequent levels of BFS neighbors are detailed in separate sections and subsections, akin to a detailed outline:

1. **Root**: Start with a comprehensive sentence that describes node $s$ and its immediate connections: "`[ROOT s] Node` $s$ `has three connections.` $s$ `is connected to` $v_1$ `via` $d_{s1}$`,` $s$ `is connected to` $v_2$ `via` $d_{s2}$`, and` $s$ `is connected to` $v_3$ `via` $d_{s3}$", where $d_{s1}, d_{s2}$, and $d_{s3}$ are textual descriptions on edges $e_{s1}, e_{s2}$, and $e_{s3}$.

2. **First-hop Neighbors**: For each first-hop neighbor of $s$, we provide a section detailing its connections: "`[1]. Node` $v_1$ `has two connections.` $v_1$ `is connected to` $v_{11}$ `via` $d_{11}$`, and` $v_1$ `is connected to` $v_{12}$ `via` $d_{12}$"; "`[2]. Node` $v_2$ `has one connection.` $v_2$ `is connected to` $v_{21}$ `via` $d_{21}$"; and "`[3]. Node` $v_3$ `has two connections.` $v_3$ `is connected to` $v_{31}$ `via` $d_{31}$`, ...`".

3. **Second (and following)-hop Neighbors**: Subsections under each first-hop neighbor detail further connections: "`[1.1]` $v_{11}$ `is connected to ...`" "`[1.2]` $v_{12}$ `is connected to...`", etc.

This structure is recursively applied up to $L$ levels deep, ensuring each node's direct connections are thoroughly described, capturing the intricate topology of the graph. The notation "`[X]`" refers to earlier parts of the summary where the connected node was initially described, aiding in understanding the network's connectivity beyond a simple hierarchical structure.

**Hidden Edges.** The neighborhood of $s$ and $t$ may not always form tree structures. As shown in Figure 3, node $v_{122}$ is not only the child node of $v_{12}$, $v_{122}$ also links back to $v_1$ to form a triangle structure. To consider a more holistic view of the node relationships, we add an extra description to the node stating its connection to pre-existing nodes. Specifically, we introduce the hidden edge information of node $v_{122}$ in `[1.2.2]` as "`In addition, ` $v_{122}$ ` is also linked to [1]` $v_1$ ` via ...`". By letting each mention of a hidden edge direct back to the respective section "`[X]`", we aim to ensure clarity and maintain the coherence of the graph's description.

**Transition Graph Document Construction.** The unified $d_{(s,t)}$, which consists of paragraphs from $s$ and $t$, aims to not only present isolated descriptions but also to highlight the interconnected nature of $G_s$ and $G_t$. In this work, we aim to illuminate the interconnectedness between $G_s$ and $G_t$ by identifying and highlighting nodes that appear in both $G_s$ and $G_t$'s local structures. These common nodes are pivotal as they link the context of one paragraph to the other. As shown in Figure 3, the common nodes are marked with half orange and half blue. For each common node, we include a cross-reference in the text where the node is mentioned, which is done by adding a note after the section index. For example, node $v_{122}$ has a section index "`[1.2.2]`" in $s$'s paragraph, and a section index "`[1.2.1]`" in $t$'s paragraph. We then add (`[1.2.1] in Paragraph` $t$) after the section index of $v_{121}$ in $s$'s paragraph. We conduct the cross-reference in the other paragraph reversely.

In practice, we keep the depth $L$ to be half of the diameter of the $G_{(s,t)}$ so that $G_s$ and $G_t$ can each cover their close neighbor information as well as an adequate number of common nodes. We further enhance the document's coherence by adding an introduction to the start of the document. Finally, the generated $d_{(s,t)}$ can be viewed as a structured document. More details can be found in Figure 3.

### 3.4 REPRESENTATION LEARNING FOR TRANSITION GRAPH NEURAL NETWORK

After obtaining a document $d_{(s,t)}$ summarizing both topology and semantic information of $G_{(s,t)}$, scalability still poses a significant challenge for language models on large-scale graphs. To conduct link prediction on a large $G_{(s,t)}$, we need to compose many documents between node pairs with duplicated content (e.g., $d_{(s,t_1)}$ and $d_{(s,t_2)}$ may largely overlap if $t_1$ and $t_2$ are neighbors).

On the other hand, GNNs are inherently designed to process graph structures efficiently, making them a promising alternative for this task. Although LLMs may not be capable of conducting large-scale link predictions, the implicit knowledge can still be utilized to train GNNs. However, a straightforward application of GNNs faces limitations: a single GNN may not fully capture the intricate interplay between the graph's structural properties and the semantic information on nodes and edges, especially in large graphs where $G_{(s,t)}$ between two nodes $s$ and $t$ can encompass thousands of nodes due to a diameter as small as 4 (Ling et al., 2023a). Moreover, independently learning local representations for $s$ and $t$ fails to account for the multi-scale interactions crucial for accurate link prediction. Each hop in the graph can reveal different structural and semantic information—immediate neighbors contribute local properties, while nodes farther away provide broader contextual insights.

**Transition Graph Neural Network (TGNN).** To better process rich textual information on large-scale graphs efficiently, we propose TGNN and introduce a novel stratified representation learning framework to captures multi-scale interactions between target nodes $s$ and $t$ by considering different "cuts" in the transition graph $G_{(s,t)}$. Each cut is defined by a pair $(n, K - n)$, where $K$ is the diameter of the $G_{(s,t)}$ between $s$ and $t$, i.e., the longest path length between $s$ and $t$ in $G_{(s,t)}$. For each cut, TGNN encompasses two directed graph convolution processes (with shared parameters): one focuses on learning the representation of $s$ of its $n$-hop neighborhood, and the other focuses on $t$'s $(K - n)$-hop neighborhood. The TGNN update function at $n$-th layer for node $u$ is given as:

$$\mathbf{h}_u^{(n)} = f_\theta(\mathbf{h}_u^{(n-1)}, \text{AGG}(\{\mathbf{h}_v^{(n)}, e_{u \to v} : v \in \mathcal{N}(u)\})), \tag{1}$$

where $v$ is the child node of $u$, $\theta$ is the parameter of the TGNN update function, $\text{AGG}(\cdot)$ is the aggregation function of TGNN. By calculating Equation (1) for $n = 1, 2, \cdots, K - 1$, we will obtain all the embeddings of $s$ and $t$ for all the cuts, namely $\{(\mathbf{h}_s^{(n)}, \mathbf{h}_t^{(K-n)})\}$, $n = \{1, 2, \cdots, K - 1\}$.

**Accelerating Transition Graph Representation Learning by Deduplicating TGNN Computation.** However, naively implementing all $K$ cuts would necessitate $2K$ times the calculation of the whole TGNN on the entire graph, resulting in duplicated computation which will be prohibitive, especially for non-small graphs that are common in the real world. To mitigate this, we exploit the hierarchical nature of our transition graph to re-order the message-passing process as a cascading process from higher-hop neighbors of $s$ (or $t$) progressively to the lower and lower layers. The TGNN update

function of cut $n$ for node $s$ given as:

$$
\begin{aligned}
\mathbf{h}^{(n)}_{u \in \mathcal{L}_n} &= \mathbf{x}_u, \\
\mathbf{h}^{(n)}_{u \in \mathcal{L}_{n-1}} &= f_\theta(\mathbf{h}^{(n-1)}_{u \in \mathcal{L}_{n-1}}, \mathrm{AGG}(\{\mathbf{h}^{(n)}_v, e_{u \to v} : v \in \mathcal{N}_{\mathcal{L}_n}(u)\})), \\
\mathbf{h}^{(n)}_{u \in \mathcal{L}_{n-2}} &= f_\theta(\mathbf{h}^{(n-2)}_{u \in \mathcal{L}_{n-2}}, \mathrm{AGG}(\{\mathbf{h}^{(n-1)}_v, e_{u \to v} : v \in \mathcal{N}_{\mathcal{L}_{n-1}}(u)\})), \\
&\vdots \\
\mathbf{h}^{(n)}_s &= f_\theta(\mathbf{h}^{(n-1)}_s, \mathrm{AGG}(\{\mathbf{h}^{(n)}_v, e_{u \to v} : v \in \mathcal{N}(u)\})),
\end{aligned}
\tag{2}
$$

where $v$ and $u$ are nodes in the transition graph $G_{(s,t)}$ and $\mathcal{L}_n$ means all the $n$-hop neighbors of $s$. The computation of $\mathbf{h}^{(n)}_s$ follows the cascading order that aggregates higher-hop neighbors' information first, as shown in Eq. (2). Since the cascading (a.k.a, Eq. (2)) for cut $n$ actually uses the node embeddings calculated in cut $n-1$, we, therefore, propose to calculate the cascading processes in a sequential order $n = 1, 2, \cdots, K-1$ so the cut $n-1$'s computation is naturally used for cut $n$, saving the later from re-compute. Therefore, the entire calculation for all the $K$ cuts for $s$ is the same as one calling of TGNN and hence we accelerate the compute by $K$ times.

**Aligning Topology Representation of $G_{(s,t)}$ with Semantic Information.** To bridge the gap between the semantic understanding capabilities of LLMs and the structural learning strengths of the TGNN, in this work, we leverage LLMs to generate embeddings $\tilde{\mathbf{h}}_{(s,t)}$ of $d_{(s,t)}$ to guide the training of TGNN in a self-supervised learning manner:

$$
\tilde{\mathbf{h}}_{(s,t)} = f_{LM}\left(d_{(s,t)}\right), \; \mathbf{h}_{(s,t)} = g(\bar{\mathbf{h}}_s \oplus \bar{\mathbf{h}}_t), \; \bar{\mathbf{h}}_s = \frac{1}{K-1} \sum_{n=1}^{K-1} \mathbf{h}^{(n)}_s, \; \bar{\mathbf{h}}_t = \frac{1}{K-1} \sum_{n=1}^{K-1} \mathbf{h}^{(n)}_t
$$

where $\oplus$ denotes embedding concatenation, $f_{LM}(\cdot)$ denote the LLM query function. We transform text on nodes and edges with pre-trained language models (e.g., LLaMA models or OpenAI's Embedding Models) and feed these attributes along with adjacency matrix of $G_s$ and $G_t$ to GNNs. The outputs are then concatenated and transformed into LLM's embedding space by a projection function $g(\cdot)$ with non-linear transformation. We seed to align the latent embeddings $\tilde{\mathbf{h}}_{(s,t)}$ produced by the LLM with the embeddings $\mathbf{h}_{(s,t)}$ generated by the GNN:

$$
\ell_{KD} = -\mathbb{E}\left[\log \frac{\exp\left(\mathrm{sim}(\tilde{\mathbf{h}}_{(s,t)}, \mathbf{h}_{(s,t)})/\tau\right)}{\sum_{k=1}^{K} \exp\left(\mathrm{sim}(\mathbf{h}_{(s,t)}, \mathbf{h}_{(s,h)})/\tau\right)}\right],
\tag{3}
$$

where the objective function is based on temperature-scaled cross-entropy loss (NT-Xent) (Chen et al., 2020) to enforce the agreement between $\tilde{\mathbf{h}}_{(s,t)}$ and $\mathbf{h}_{(s,t)}$ compared with latent embedding $\mathbf{h}_{(s,h)}$ from negative pairs. Furthermore, to calibrate $\mathbf{h}_{(s,t)}$ more towards the link prediction (and edge classification) task, we incorporate standard binary cross-entropy loss $\ell_{LP}$ for tuning GNNs. Note that for dealing with highly imbalanced label distribution for edge classification tasks, we use weighted cross-entropy loss (e.g., Focal Loss (Lin et al., 2017)) instead.

Finally, the overall objective of the LLM-enhanced Representation Learning for predicting links on edge-attributed graphs is written as $\ell = \lambda_1 \ell_{KD} + \lambda_2 \ell_{LP}$, where $\lambda_1$ and $\lambda_2$ are hyperparameters.

**Complexity Analysis.** Given the transition graph $G_{(s,t)}$ with the diameter $K$, we use Breadth-first Search (time complexity $O((N+E)/2)$ (BFS) with the depth $K/2$ to extract $G_s$ and $G_t$, respectively. We then use pre-order traversal to obtain the document $d_{(s,t)}$, which leads the total complexity to be $O(N + E + N)$. The time complexity for Pre-trained LMs to process $d_{(s,t)}$ is $O(P^2)$, where $P$ denotes the number of tokens in $d_{(s,t)}$. Moreover, the GNN module requires $O(|E| \cdot f + N^2)$, where $f$ denotes the dimension of the embedding on nodes/edges. Overall, The complexity encapsulates the stages of BFS tree construction, document processing with Transformers, and GNN learning, which gives the training time complexity $O(2N + E + P^2 + |E| \cdot f + N^2)$. However, during the inference stage, the complexity of our work simply reduces to the complexity of normal GNNs: $O(|E| \cdot f + N^2)$ as we do not need to construct documents during the inference phase.

## 4 EXPERIMENT

**Setup.** This paper focuses on link prediction on TEGs, which aims to predict whether there will be a strong connection between two nodes in the adopted datasets based on their transition graph. We run experiments on five real-world networks: Amazon-Movie (He & McAuley, 2016), Amazon-Apps (He & McAuley, 2016), GoodReads-Children (Wan et al., 2019), GoodReads-Crime (Wan et al., 2019), and StackOverflow. More specific dataset statistics can be found in the Appendix A.3. We evaluate the performance using four standard metrics: Mean Reciprocal Rank (MRR), Normalized Discounted Cumulative Gain (NDCG), Area Under ROC Curve (AUC) metric, and F1 score.

**Comparison Methods.** We compare our model with *general GNNs*, *language model integrated GNNs*, and *large language models*. For *general GNNs*, we select MeanSAGE (Hamilton et al., 2017), MaxSAGE (Hamilton et al., 2017), GIN (Xu et al., 2019) and RevGAT (Li et al., 2021), which only use an adjacency matrix as the input. For *language model-enhanced GNNs*, we utilize Pre-trained LMs, e.g., BERT (Devlin et al., 2019), to acquire text representations on edges. Our baselines consist of BERT + Graph Transformer (GTN) (Yun et al., 2019), BERT + GINEConv (Hu et al., 2019) and BERT + EdgeConv (Wang et al., 2019). Furthermore, we also incorporate state-of-the-art edge-aware GNN - Edgeformer (Jin et al., 2022), which is constructed based on graph-enhanced Transformers to combine language modeling into each layer of the Graph transformer. A novel graph foundation model - THLM (Zou et al., 2023) is also included that integrates language modeling with GNN training. Finally, we adopt state-of-the-art LLMs, i.e., LLAMA-3-70B and GPT-4O by directly translating the transition graph $G_{(s,t)}$ between the node pair $(s,t)$ to natural language as (Fatemi et al., 2023) do. Note that state-of-the-art *language model integrated GNNs*, namely EdgeFormer, cannot incorporate text on nodes. For a fair comparison, we present two variants of our method: LINK2DOC does not consider node texts, and LINK2DOC-NT takes node text into account.

**Implementation Details.** To process all node and edge text, we leverage OpenAI's embedding model[1] with dimension $3,072$. For both *general GNNs* and *language model enhanced GNNs*, the dimensions of the initial node and edge embeddings are further normalized to $64$ and $128$ respectively. Additionally, for the Edgeformer model, we adhere to the same experimental settings as outlined in (Jin et al., 2022). Our model uses Graph Transformer (GTN) as our backbone in Eq. (1), where both node and edge embeddings mirror those of *language model integrated GNNs*. For LINK2DOC, we follow the same settings as general GNNs to obtain node and edge embeddings from OpenAI's embedding model. We set $\lambda_1 = 1$ and $\lambda_2 = 2$ respectively. All GNN baseline layers are set to 2. The temperature $\tau$ in Eq. (3) to 2. We use Adam as the optimizer with a learning rate of $1e-5$. The batch size is $1,024$. We run our model and other baselines 10 times with different random seeds and report the average performance.

### 4.1 RESULTS ON LINK PREDICTION

As can be seen from Table 1 and Table 2, LINK2DOC can consistently achieve better performance than other methods. Specifically, LINK2DOC outperforms the second best on average 5% of both AUC and F1 (Table 1) and 10% of both MRR and NDCG across all datasets (Table 2). We further draw several observations from the results. *1) There are no clear differences between general GNNs and edge-aware GNNs*: both types of GNNs show comparable performance, with edge-aware GNNs having a slight edge but not consistently outperforming general GNNs in all metrics, which implies co-training language models with GNNs still cannot capture the subtle cues on edge texts. For instance, while EDGEFORMER achieves the second-best AUC on the Goodreads-Children dataset, it doesn't consistently outperform general GNNs like MAXSAGE across all datasets. *2) Directly summarizing topology and letting LLMs make predictions may not perform well*: LLMs, such as LLAMA-3-70B and GPT-4O, tend to underperform compared to specialized GNNs and our proposed LINK2DOC. It is evident that state-of-the-art LLMs may not be able to fully understand graph topology from the linear topology summarization, highlighting the advantage of the composed document. *3) Text on nodes can further improve the performance:* As can be seen from the table, even though LINK2DOC can achieve a generally better performance than other approaches, by considering the text on nodes, LINK2DOC-NT can achieve a generally better performance than its no node-text version.

---

[1]https://platform.openai.com/docs/guides/embeddings/embedding-models

| | Goodreads-Children | | Goodreads-Crime | | Amazon-Apps | | Amazon-Movie | | StackOverflow | |
|---|---|---|---|---|---|---|---|---|---|---|
| | AUC | F1 | AUC | F1 | AUC | F1 | AUC | F1 | AUC | F1 |
| **General GNN** | | | | | | | | | | |
| MAXSAGE | 0.870 | 0.637 | 0.858 | 0.624 | 0.727 | 0.571 | 0.706 | 0.527 | 0.895 | 0.631 |
| MEANSAGE | 0.828 | 0.611 | 0.829 | 0.615 | 0.700 | 0.569 | 0.686 | 0.525 | 0.887 | 0.615 |
| REVGAT | 0.862 | 0.622 | 0.839 | 0.619 | 0.662 | 0.562 | 0.689 | 0.541 | 0.819 | 0.533 |
| GIN | 0.859 | 0.571 | 0.857 | 0.577 | 0.705 | 0.543 | 0.692 | 0.512 | 0.873 | 0.605 |
| **LM-enhanced GNN** | | | | | | | | | | |
| GTN | 0.880 | 0.654 | 0.863 | 0.640 | 0.728 | 0.572 | 0.742 | 0.539 | 0.911 | 0.675 |
| GINECONV | 0.881 | 0.657 | 0.864 | 0.636 | 0.701 | 0.573 | 0.692 | 0.543 | 0.920 | 0.681 |
| EDGECONV | 0.879 | 0.646 | 0.860 | 0.622 | 0.692 | 0.551 | 0.682 | 0.532 | 0.835 | 0.563 |
| THLM | 0.871 | 0.651 | 0.871 | 0.635 | 0.718 | 0.587 | 0.749 | 0.534 | 0.911 | 0.659 |
| EDGEFORMER | 0.882 | 0.662 | 0.862 | 0.643 | 0.722 | 0.580 | 0.744 | 0.540 | 0.903 | 0.663 |
| **LLMs** | | | | | | | | | | |
| LLAMA-3-70B | 0.832 | 0.573 | 0.869 | 0.587 | 0.694 | 0.509 | 0.643 | 0.482 | 0.252 | 0.471 |
| GPT-4O | 0.878 | 0.609 | 0.889 | 0.604 | 0.712 | 0.512 | 0.659 | 0.503 | 0.407 | 0.561 |
| LINK2DOC | **0.902** | **0.705** | **0.901** | **0.652** | **0.762** | **0.588** | **0.753** | **0.553** | **0.938** | **0.697** |
| LINK2DOC-NT | – | – | – | – | **0.769** | **0.595** | **0.759** | **0.565** | **0.940** | **0.707** |

Table 1: The performance comparison of Link Prediction on all datasets (the higher the better), where the bests are highlighted with **bold**, and the second bests are highlighted with underline. Note that − indicates the dataset does not have text on nodes so that LINK2DOC-NT cannot be conducted.

| | Goodreads-Children | | Goodreads-Crime | | Amazon-Apps | | Amazon-Movie | | StackOverflow | |
|---|---|---|---|---|---|---|---|---|---|---|
| | MRR | NDCG | MRR | NDCG | MRR | NDCG | MRR | NDCG | MRR | NDCG |
| **General GNN** | | | | | | | | | | |
| MAXSAGE | 0.2059 | 0.3342 | 0.2130 | 0.3372 | 0.2119 | 0.3938 | 0.2148 | 0.4299 | 0.2256 | 0.3313 |
| MEANSAGE | 0.2156 | 0.3619 | 0.2006 | 0.3199 | 0.2179 | 0.3951 | 0.2433 | 0.4340 | 0.2155 | 0.3351 |
| REVGAT | 0.2079 | 0.3567 | 0.1921 | 0.2997 | 0.2039 | 0.3865 | 0.2253 | 0.4318 | 0.2159 | 0.3369 |
| GIN | 0.2147 | 0.4160 | 0.2354 | 0.3644 | 0.2313 | 0.3486 | 0.2061 | 0.4305 | 0.2254 | 0.3351 |
| **LM-enhanced GNN** | | | | | | | | | | |
| GTN | 0.2239 | 0.4207 | 0.2536 | 0.4398 | 0.3134 | 0.4296 | 0.2872 | 0.4958 | 0.2321 | 0.4201 |
| GINECONV | 0.2458 | 0.4399 | 0.2628 | 0.4629 | 0.2916 | 0.4467 | 0.2587 | 0.4472 | 0.2340 | 0.4243 |
| EDGECONV | 0.2389 | 0.4281 | 0.2486 | 0.4265 | 0.2871 | 0.4318 | 0.2492 | 0.4432 | 0.2326 | 0.4218 |
| THLM | 0.1732 | 0.2998 | 0.2416 | 0.3964 | 0.2337 | 0.3845 | 0.2969 | 0.4284 | 0.1696 | 0.3283 |
| EDGEFORMER | 0.1754 | 0.3000 | 0.2395 | 0.3875 | 0.2239 | 0.3771 | 0.2919 | 0.4344 | 0.1754 | 0.3339 |
| **LLMs** | | | | | | | | | | |
| LLAMA-3-70B | 0.1356 | 0.2127 | 0.0692 | 0.0778 | 0.0500 | 0.1692 | 0.0683 | 0.1657 | 0.1421 | 0.2144 |
| GPT-4O | 0.2079 | 0.4106 | 0.2684 | 0.3633 | 0.2740 | 0.3697 | 0.2352 | 0.4228 | 0.2299 | 0.3751 |
| LINK2DOC | **0.3167** | **0.5988** | **0.3518** | **0.6115** | **0.4139** | **0.5287** | **0.3926** | **0.6141** | **0.3618** | **0.6359** |

Table 2: The performance comparison of Link Prediction on all datasets (the higher the better), where the bests are highlighted with **bold**, and the second bests are highlighted with underline.

## 4.2 RESULTS ON EDGE CLASSIFICATION

In the task of edge classification, the model is asked to predict the category of each edge based on its associated text and local network structure. There are 5 categories for edges in the Amazon-APPs dataset (i.e., from 1 star to 5 star). The results of the 5-class edge-type classification are shown in Table 3. As clearly can be observed in the table, LINK2DOC improves the AUC by an average of 5% and the F1 score by 4.2% compared to other models, demonstrating its superior performance in edge classification.

| | Amazon-APPs | |
|---|---|---|
| | AUC | F1 |
| MEAN-SAGE | 0.551 | 0.479 |
| GINE | 0.573 | 0.488 |
| GTN | 0.567 | 0.503 |
| EDGEFORMER | 0.612 | 0.526 |
| LINK2DOC | **0.626** | **0.541** |

Table 3: Comparison on Edge Classification on Amazon-APPs dataset.

## 4.3 ABLATION STUDY AND PARAMETER ANALYSIS

We further demonstrate the effectiveness of each component in our framework and analyze the importance of different hyper-parameter settings.

**Performance Elevation from Transition Document.** We first aim to check the general performance elevation brought by the composed transition document $d_{(s,t)}$. We adopt three GNNs using BERT to obtain embeddings on edges and illustrate whether using the composed $d_{(s,t)}$ as a reference would improve their performance in both link prediction and edge classification tasks. The results on the

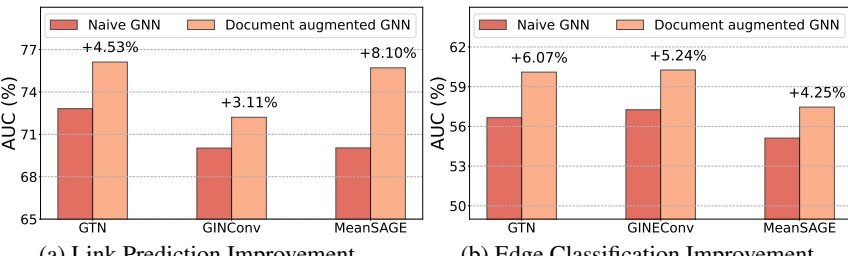

(a) Link Prediction Improvement  (b) Edge Classification Improvement

Figure 4: Leveraging composed documents to enhance base GNNs on Amazon-APPs dataset.

Amazon-APPs dataset are presented in Figure 4. As can be seen from the figure, the document-augmented GNNs excel in their general version with an average improvement of $5\%$. By summarizing all relations from $s$ to $t$ as a coherent document, language models can give positive feedback on the proposed self-supervised learning module to guide different GNNs in learning. An additional ablation study on the effectiveness of TGNN is provided in Appendix A.4.

**Hyperparameter Analysis.** We then aim to investigate the sensitivity of the key hyperparameter $\lambda_2$ and their impact on LINK2DOC's performance. Specifically, since $\lambda_1$ controls the basic objective function for link prediction, we fix $\lambda_1 = 1$ and show the link prediction performance on the Amazon-APPs dataset under different $\lambda_2$ values (ranging from $0$ to $5$). As shown in Figure 5, both metrics show consistent results across varying parameter values. By comparing with the second-best methods (highlighted with red dash horizontal lines), LINK2DOC with various $\lambda_2$ values can achieve overall better results. This demonstrates that our model maintains superior performance across different configurations, highlighting its stability and effectiveness.

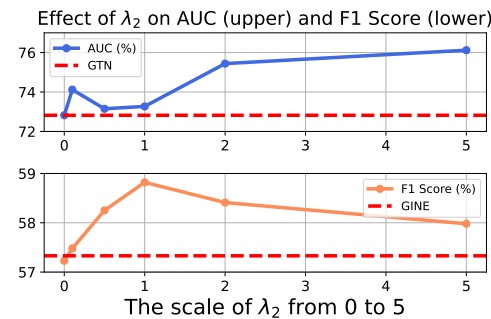

Figure 5: The performance on Amazon-APPs.

### 4.4 RUNTIME ANALYSIS

We further illustrate the runtime comparison between our proposed LINK2DOC with the state-of-the-art competitor - Edgeformer. The comparison table (Table 4) highlights the efficiency of our method, Link2Doc, which significantly outperforms Edgeformer in both inference and training times. For the Goodreads-Children dataset, Link2Doc is approximately 29 times faster in inference and more than twice as fast in training. On the Goodreads-Crime dataset,

|  | Children | Crime |
|---|---|---|
| **Inference Time (s)** | | |
| EDGEFORMER | 1450.67 | 3307.66 |
| LINK2DOC | 49.35 | 23 |
| **Training Time (h)** | | |
| EDGEFORMER | 12.17 | 12.65 |
| LINK2DOC | 5.253 | 7.04 |

Table 4: Comparison of inference and training time on Goodreads-Children and Goodreads-Crime.

Link2Doc demonstrates an even greater advantage, being about 144 times faster in inference and almost twice as fast in training. These improvements stem from Link2Doc's design, which does not require fine-tuning language models; instead, it builds document representations and uses a pre-trained GNN, avoiding the extensive matrix calculations in Edgeformer's cross-attention design. Consequently, Link2Doc is more scalable and efficient for large-scale link prediction tasks.

## 5 CONCLUSION

In this work, we study the problem of link prediction on textual-edge graphs, where existing GNN-based and LLM-based methods may fall short of jointly capturing both semantic and topology information to make more accurate link predictions. We present a novel framework LINK2DOC that learns and aligns the semantic representation and topology representation by 1) building a structured document to preserve both topology and semantic information; 2) proposing a Transition Graph Neural Network module for better learning representations of the transition graph; and 3) designing a self-supervised learning module to let TGNN have text understanding ability like LLMs. Our method generally outperforms other approaches from multiple aspects on five datasets.

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

## A  Supplemental Material

### A.1  Algorithm of Transition Document Composition

---

**Algorithm 1:** Transition Document Composition

---

**Data:** The transition graph $G_{(s,t)}$, the diameter $K$ of $G_{(s,t)}$.

**Result:** Composed document $d_{(s,t)}$ with hierarchical relation, hidden edge references, and cross-paragraph references.

1. $G_s \leftarrow BFS(\text{ROOT} = s, \text{GRAPH} = G_{(s,t)}, \text{DEPTH} = K//2)$;

2. $G_t \leftarrow BFS(\text{ROOT} = t, \text{GRAPH} = G_{(s,t)}, \text{DEPTH} = K//2)$;

                `/* Obtaining local structure of s and t' neighbor by breadth-first search with depth K//2. */`

3. $E_s^{\text{hidden}} \leftarrow \{e_{ij} | \forall v_i \in G_s, v_j \in G_s, e_{ij} \in G_{(s,t)}, e_{ij} \notin G_s\}$;

4. $E_t^{\text{hidden}} \leftarrow \{e_{ij} | \forall v_i \in G_t, v_j \in G_t, e_{ij} \in G_{(s,t)}, e_{ij} \notin G_t\}$;

    `/* For both subgraphs $G_s$ and $G_t$, we obtain hidden edges. */`

5. $V^{\text{cross}} \leftarrow V_s \cup V_t$;

                   `/* Get common nodes shared by $G_s$ and $G_t$. */`

6. Initiate the document $d_{(i,j)}$ with an initial prompt: `""We have two paragraphs that summarize the relation between s and t..."`;

7. **for** *each node $v_i \in G_s$* **do**

    ⌊ 8. Assign document sections (e.g., [SEC. 1.1]) following pre-order traversal of $G_s$;

9. Assign hidden edge following $E_s^{\text{hidden}}$ to $d_{(i,j)}$ ;

10. **for** *each node $v_i \in G_t$* **do**

    ⌊ 11. Assign document sections (e.g., [SEC. 1.1]) following pre-order traversal of $G_t$;

12. Assign hidden edges following $E_t^{\text{hidden}}$ to $d_{(i,j)}$ ;

13. Assign cross-paragraph reference $\forall v_i \in V^{\text{cross}}$;

---

We provide the full procedure of producing the document $d_{(i,j)}$ based on the node pair's transition graph $G_{(s,t)}$ in Algorithm 1. From Line 1-2, we obtain the respective local structure of $s$ and $t$ based on their transition graph $G_{(s,t)}$ by BFS search. From Line 3-4, for both $G_s$ and $G_t$, we extract hidden edges that are not covered in the BFS tree. In Line 5, we get the intersection of the node sets $V_s$ and $V_t$ for recording the cross-paragraph nodes. Next, we initiate the document with the initial prompt and traverse each node in $G_s$ and $G_t$ to assign document section indices and relations to the document. Finally, we add hidden edges and cross-paragraph references as denoted in Line 12-13.

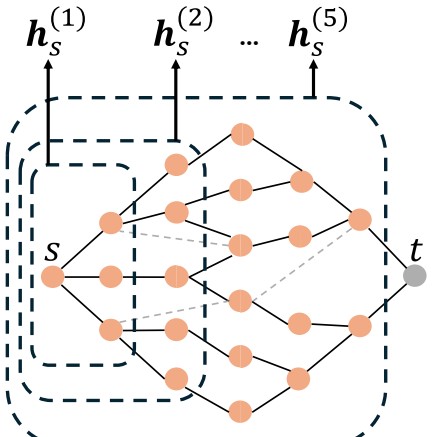

Figure 6: The stratified representation learning of Transition Graph Neural Network.

## A.2 TRANSITION GRAPH NEURAL NETWORK

Considering a $(K-1)$-layer graph neural network, TGNN returns the embedding of $s$ at each embedding updating layer of the GNN, as shown in Figure 6. The strategy effectively minimizes duplicated computation by reusing convolution outputs across different cuts, resulting in a total computational cost equivalent to an $K$-layer GNN. While it shares computational complexity with a standard $K$-layer GNN, it is not strictly equivalent in terms of representation learning. By capturing and utilizing multi-scale representations at each layer $n$, the proposed approach offers potential advantages in expressiveness and performance for link prediction tasks.

## A.3 DATASETS

**Data.** We run experiments on five real-world networks: Amazon-Movie (He & McAuley, 2016), Amazon-Apps (He & McAuley, 2016), GoodReads-Children (Wan et al., 2019), GoodReads-Crime (Wan et al., 2019), and StackOverflow[2]. Amazon is a user-item interaction network, with reviews serving as textual content associated with the edges. Goodreads is a reader-book network, that utilizes readers' comments as textual information within the edges. StackOverflow is an expert-question network, and there will be an edge when an expert posts an answer to a question. The statistics of the four datasets can be found in Table 5.

Table 5: Dataset Statistics

| Dataset | # Node | # Edge |
|---|---|---|
| Goodreads-Children | 192,036 | 734,640 |
| Goodreads-Crime | 385,203 | 1,849,236 |
| Amazon-Apps | 100,468 | 752,937 |
| Amazon-Movie | 173,986 | 1,697,533 |
| Stack OverFlow | 129,322 | 281,657 |

## A.4 ABLATION STUDY ON THE EFFECTIVENESS OF TRANSITION GRAPH NEURAL NETWORK

| | Amazon-Apps | | Amazon-Movie | | Goodreads-Children | | Goodreads-Crime | | StackOverflow | |
|---|---|---|---|---|---|---|---|---|---|---|
| | AUC | F1 | AUC | F1 | AUC | F1 | AUC | F1 | AUC | F1 |
| **Single-Cut** | 0.7620 | 0.5880 | 0.7530 | 0.5530 | 0.9020 | 0.7050 | 0.9010 | 0.6520 | 0.9185 | 0.6841 |
| **Multi-Cut** | **0.7697** | **0.5997** | **0.7731** | **0.5755** | **0.9146** | **0.7099** | **0.9124** | **0.6661** | **0.9374** | **0.6968** |

Table 6: Ablation study comparison between LINK2DOC with single cut versus multi-cut across five datasets, where the best values are bolded.

---

[2]https://www.kaggle.com/datasets/stackoverflow/stackoverflow

We further demonstrate the effectiveness of considering multiple cuts for learning a better representation of the transition graph. As can be seen from Table 6, Single-Cut denotes we only split $G_{(s,t)}$ in half, where each $G_s$ and $G_t$ have the depth of $K/2$.

In general, the Multi-Cut strategy consistently outperforms the Single-Cut approach across all datasets. For example, in the Amazon-Apps dataset, the Multi-Cut method achieves an AUC of $0.7697$ compared to $0.7620$ with the Single-Cut method, and an F1 score of $0.5997$ compared to $0.5880$. This trend is observed across other datasets as well, such as Goodreads-Crime, where the Multi-Cut approach results in an AUC of $0.9124$ versus $0.9010$ for Single-Cut, and an F1 score of $0.6661$ compared to $0.6520$. The improvement is particularly notable in the StackOverflow dataset, where the AUC increases from $0.9185$ with Single-Cut to $0.9374$ with Multi-Cut, and the F1 score rises from $0.6841$ to $0.6968$.

Overall, the results clearly indicate that the Multi-Cut strategy leads to better performance in both AUC and F1 scores, suggesting that the model benefits from the multi-scale representation learning provided by the Multi-Cut approach. This likely enhances the model's ability to capture more comprehensive and diverse neighborhood information, leading to improved prediction accuracy.

## A.5 LIMITATIONS

This work relies on pre-trained language models like GPT models and LLAMA models, which may introduce potential biases in the model's understanding of text. These biases can be perpetuated in the graph representations and impact the fairness and accuracy of link predictions.

