# OpenReview forum: "Link Prediction on Textual Edge Graphs"
_ICLR.cc/2025/Conference — Submitted to ICLR 2025_

### Official Review · Reviewer_JRVL · 2024-10-16

**Soundness:** 3
**Presentation:** 3
**Contribution:** 3
**Rating:** 5
**Confidence:** 3

**Summary:**

The paper introduces LINK2DOC, a novel framework designed for link prediction on Textual-edge Graphs (TEGs), which are graphs characterized by rich text annotations on edges. The framework aims to address the limitations of existing methods by capturing both the graph topology and the semantic information embedded in the text on the edges. LINK2DOC achieves this by summarizing neighborhood information between node pairs into coherent documents, which preserve both semantic and topological relations. Additionally, it employs a stratified GNN architecture to capture multi-scale interactions between nodes and utilizes a self-supervised learning module to enhance the GNN’s ability to understand and process textual information. Extensive experiments demonstrate that LINK2DOC outperforms existing edge-aware GNNs and language models on multiple real-world datasets.

**Strengths:**

1. LINK2DOC introduces a novel approach to link prediction by converting local graph information into coherent documents, effectively combining both graph topology and rich semantic information on edges. This offers a unique solution to the challenges of learning representations on TEGs.

2. The proposed specialized Transition Graph Neural Network captures multi-scale interactions between target nodes in a stratified manner, improving the model's ability to process complex topological relationships and edge semantics at different levels of granularity.

3. The framework is rigorously tested on four real-world datasets, showing superior performance over general GNNs and competitive results against edge-aware GNNs. The inclusion of link prediction, edge classification, and runtime analysis further demonstrates the robustness and practical applicability of LINK2DOC.

**Weaknesses:**

1. When consider about the time complexity, the LLM encoder part is not taken into consideration.
2. Some baseline is lack of consideration. One advantage of the text-attributed graph is that it can be naturally represented as the textual representation. The potential reason why LLMs cannot do well are two-fold. (i) the annotation error or labeling bias. (which I think case study is important to show whether the LLM prediction is reasonable) (ii) out-of-distribution issue. A smaller LM, e.g., sentencebert can achieve better performance
3. For the GNN baseline, GLEM baseline which train both bert and GNN is not taken into consideration[1]
4. The proposed method shows certain similarity with the clip model, I think the model may benefit more from large-scale pre-training. Moreover, the model may have good zero-shot generalization capability, which I would suggest the author to check more on it.

[1] Zhao, Jianan, et al. "Learning on Large-scale Text-attributed Graphs via Variational Inference." The Eleventh International Conference on Learning Representations.

**Questions:**

1. Could labeling bias or annotation errors be affecting the LLM's performance, and would a case study help demonstrate if the LLM predictions are reasonable?
2. Given the similarities between the proposed method and models like CLIP, do you think large-scale pre-training could improve performance, and have you considered exploring the model's potential zero-shot generalization capabilities?

---

### Official Review · Reviewer_LHgP · 2024-10-26

**Soundness:** 3
**Presentation:** 3
**Contribution:** 3
**Rating:** 5
**Confidence:** 4

**Summary:**

This paper investigates representation learning at the edge level on a specific type of text-attributed graph where edges carry semantic information. To address the limitations of previous methods in capturing contextualized edge semantics and the graph's topology, they propose Link2Doc. This approach summarizes the semantic and structural information of the neighborhood as a document. They then perform representation learning using a multi-scale Graph Neural Network (GNN) supplemented with an alignment module. This method demonstrates state-of-the-art performance across a variety of edge prediction tasks.

**Strengths:**

1. This paper studies an interesting research question with clear motivation. The example given in Figure 1 well demonstrates the importance of edge semantics in citation networks.
2. The method proposed in this paper seems sound.
3. The experimental results are promising.

**Weaknesses:**

1. Edges with semantic information have been widely studied in some other areas, like recommendation systems and multi-tabular prediction [1], authors may add some discussions on these related areas.
2. Despite the promising performance, the methods proposed in this paper have high module complexity. Using LLM to summarize neighbor information is not innovative, and encoding long text can be costly in terms of time and money. In the experiment section, I didn't see any ablation study on the self-supervised learning-based alignment, nor any analysis of the time and money cost of generating embeddings with LLM. Moreover, I think the time comparison between Link2Doc and Edgeformer is not very meaningful, as it's difficult to evaluate the API call time to OpenAI.
3. The modules present in this paper are relevant to [2,3], but they are not discussed in this paper.
4. In terms of performance comparison, I think parts of the improvement are brought by more powerful text encoders (openai embedding model with more context length), which limits the applicability of this model in real industrial scenarios (like [1]).

[1] Wang, Minjie, et al. "4DBInfer: A 4D Benchmarking Toolbox for Graph-Centric Predictive Modeling on Relational DBs." arXiv preprint arXiv:2404.18209 (2024).
[2] Wen, Zhihao, and Yuan Fang. "Augmenting low-resource text classification with graph-grounded pre-training and prompting." Proceedings of the 46th International ACM SIGIR Conference on Research and Development in Information Retrieval. 2023.
[3] Song, Yunchong, et al. "Ordered gnn: Ordering message passing to deal with heterophily and over-smoothing." arXiv preprint arXiv:2302.01524 (2023).

**Questions:**

1. May you demonstrate the cost of using openai models?
2. Could you summarize the whole process (not the document construction) using the format of an algorithm?
3. May this framework be extended to solve other problems like heterophily in node classification by considering the semantics of edges?

---

### Official Review · Reviewer_Xi2w · 2024-11-04

**Soundness:** 3
**Presentation:** 2
**Contribution:** 2
**Rating:** 5
**Confidence:** 4

**Summary:**

This paper addresses the problem of link prediction on textual-edge graphs, highlighting that existing GNN-based and LLM-based methods often fail to capture both semantic and topological information effectively. To overcome these limitations, the authors propose LINK2DOC, a novel framework that constructs a structured document to preserve this information and employs a Transition Graph Neural Network (TGNN) for enhanced representation learning. Additionally, incorporates a self-supervised learning module that aligns the semantic understanding of TGNN with the capabilities of LLMs. Experiments across five datasets demonstrate that LINK2DOC consistently outperforms existing methods in link prediction accuracy.

**Strengths:**

This paper presents an innovative approach to address the shortcomings of existing GNN-based and LLM-based methods in link prediction. The authors effectively highlight the limitations of current methodologies and introduce a novel framework that integrates both graph structure and textual information.

**Weaknesses:**

One notable limitation is that the paper’s Transition Graph Document Construction approach appears to be similar with the Graph2Text component from the TAGA framework, as introduced in TAGA: Text-Attributed Graph Self-Supervised Learning by Synergizing Graph and Text Mutual Transformations https://arxiv.org/abs/2405.16800. If that is your work, please cite this paper properly, and discuss about the difference between your method with perspective of the algorithm development and complexity?

**Questions:**

1. In the PROBLEM FORMULATION section, you mention that linearly summarizing graph topology may cause LLMs to struggle with understanding information propagation from s to t and the contextual dependencies among nodes. However, in the ablation study, you only compare the performance differences between the “without document” and “document-augmented” settings. Could you add an ablation study comparing the performance of linearly summarizing graph topology versus using your proposed Transition Document Construction? This would help highlight the specific advantages of the Transition Document Construction approach.
2. In your experiments, the value of K used for K cuts in TGNN is not mentioned. Could you clarify what value of K was used to achieve your reported results?

**Details Of Ethics Concerns:**

this paper can be easily found in arxiv https://arxiv.org/abs/2405.16606, is it acceptable?

---

### Official Review · Reviewer_6k22 · 2024-11-05

**Soundness:** 1
**Presentation:** 1
**Contribution:** 1
**Rating:** 3
**Confidence:** 3

**Summary:**

This paper focuses on Link Prediction on Textual-edge Graphs. For current approaches, the authors challenge LM's ability to understand graph topology, and GNN's ability to consider contextual information on all connections. This paper proposes an integrated framework, Link2Doc, to jointly consider topology and semantic information in TEG, which summarizes semantic relations between nodes in text, and processes topology information with a transition GNN. This paper compares Link2Doc with SOTA methods in 4 datasets and shows some improvements.

**Strengths:**

1. The analysis of challenges in LLM-based and GNN-based methods are interesting, supported by real examples in Fig 1.
2. The proposed transition graph provides a clear view of target node pairs, avoiding information over-smoothing when propagating on whole graph.

**Weaknesses:**

1. Bad representation. The challenges of LLM-based and GNN-based methods are twisted together. Not clearly conveyed in texts and supported by experiments.
2. LM-enhanced GNNs use BERT to extract text representations, while other models use LLaMA-3 and GPT-4o, making comparison unfair. Besides, even though LM-enhanced GNNs use BERT, these models still perform the second best generally, making the improvements of Link2Doc doubtable.
3. The GNN layers are set to 2. The review suspect that, will the proposed challenges and examples still stand in such a shallow graph setting.
4. The compared baselines are not in their best performance, making overall comparison inconvincing.

**Questions:**

see above

---

### Official Review · Reviewer_o9L3 · 2024-11-05

**Soundness:** 2
**Presentation:** 2
**Contribution:** 2
**Rating:** 5
**Confidence:** 4

**Summary:**

This paper studies link prediction on textual-edge graphs (TEGs). They proposed a framework LINK2DOC, which models TEGs from two perspectives, i.e., the text view (text-of-graph) and the graph view (graph-of-text). Via performing contrastive learning to align the representations produced by the two views, they distill the knowledge from LLMs to GNNs for efficient inference.

**Strengths:**

1. The question of LP on TEGs is practical.

2. The idea of using (s,t)-transition graph for link prediction is interesting and makes sense to me.

3. Dstillinig knowledge from teacher LLM to student GNN is a well-established approach with proven effectiveness, which suits the setting of LP on TEGs.

**Weaknesses:**

1. I found it difficult to capture the key aspects of the proposed method. The authors may consider reorganizing the manuscript to put more emphasis on the overall pipeline and refer readers to appendix for details. The current version contains a lot of details and seems overwhelming to me.

2. Confusing experimental settings and missing baselines. Please see Questions.

**Questions:**

1. Regarding the experiments, I suggest the authors organize their method and baselines based on the information used: (1) do they use the node feature information? (2) do they use the graph structure information? (3) do they use the edge feature information?
The current layout makes it hard to compare the proposed method with baselines under a fair setting.

2. Similar to above, for each method, what LM was used to generate the textual embeddings? Since the capabilities of different LMs vary a lot, this information is critical for a fair judgement.

3. The authors mentioned that the texts were processed with OpenAI's embedding model with d=3072. Is this embedding used only during the contrastive learning process as the output of the text view? What is the input features for the student GNN?

4. As a paper specific to link prediction, it is weird not to include any GNN4LP baselines like NCN and BUDDY. If this setting is not compatible with GNN4LP methods, please explain the reasons.

---

### Meta-Review · Area_Chair_pC2k · 2024-12-16

**Metareview:**

Reviewers find that the problem of LP on textual edge graphs is well motivated with solid analysis and example in Figure 1. The transition graph idea is interesting.

However, experimental setup requires further clarification, and the presentation and organization of the paper can be better organized to highlight the key aspects.

**Additional Comments On Reviewer Discussion:**

The authors did not provide any response.

---

### Decision · Program_Chairs · 2025-01-22

Reject